# Place of Residence Is Associated with Dietary Intake and BMI-SDS in Children and Adolescents: Findings from the DONALD Cohort Study

**DOI:** 10.3390/ijerph21010046

**Published:** 2023-12-28

**Authors:** Janosch Klemm, Ines Perrar, Christian Borgemeister, Ute Alexy, Ute Nöthlings

**Affiliations:** 1Center for Development Research (ZEF), University of Bonn, Genscherallee 3, 53113 Bonn, Germany; cb@uni-bonn.de; 2Institute of Nutrition and Food Sciences (IEL), Nutritional Epidemiology, University of Bonn, Friedrich-Hirzebruch-Allee 7, 53115 Bonn, Germany; iperrar@uni-bonn.de (I.P.); alexy@uni-bonn.de (U.A.); noethlings@uni-bonn.de (U.N.)

**Keywords:** dietary intake, children, adolescents, urban settings, spatial trends

## Abstract

We aimed to determine whether place of residence in the German urban food environment is associated with habitual dietary intake (energy, macronutrients, and food groups) and body mass index (standard deviation score of BMI and BMI-SDS). Our hypothesis was that place of residence may explain some variation in dietary intake and nutritional outcomes. For the cross-sectional analyses of DONALD study data, we grouped participants according to their geocoded residence in the north or south of Dortmund. We applied robust multi-level mixed effects regression models using residence as a predictor and (1) BMI-SDS or (2) dietary data (daily intake of energy (kcal), macronutrients (energy percentage), or food groups (g/1000 kcal)) as the outcome. Models were adjusted for age, sex, and household socio-economic status. An analysis was carried out on 1267 anthropometric measurements collected annually from 360 participants aged 6–18 years (935 3-day weighed dietary records from 292 participants) between 2014 and 2019. In the fully adjusted models, residence in the south was associated with a lower BMI-SDS (β = −0.42, *p* = 0.02), lower intake of sugar-sweetened beverages (β = −47.00, *p* = 0.04), and higher intake of vegetables (β = 11.13, *p* = 0.04). Findings suggest that the place of residence, beyond individuals’ socio-economic statuses, may be a contributing factor to dietary quality.

## 1. Introduction

Dietary risk factors are a major cause of global illness [1,2]. Understanding the role of external factors, such as the food environment around the individual (here determined by place of residence), is important regarding dietary intake: Spatial patterns of dietary intake can be the first point of investigation to understand how variation occurs in different target groups [3,4,5] and identify potential pathways to making healthy, sustainable diets more widely consumed. School-aged children and adolescents are a critical age group for this, as they require healthy diets for proper growth and development [6]. Additionally, adult dietary patterns are shaped during childhood [7], making this a key target group for possible interventions. Existing analyses on spatial patterns have yielded strong insights into the role of the built environment on dietary intake and health outcomes [8,9,10], establishing the concepts of food environment [11,12], food deserts [13,14], and food swamps [10,15] in nutrition and food security research.

Recent evidence on the spatial variation in dietary intake has highlighted specifically the role of neighborhood socio-economic statuses for BMI and dietary intake [16,17,18,19,20,21,22], but little to no research has been carried out on this topic in Germany. It is not established whether residence and location are relevant factors for dietary intake within the German population [23]. While research is available for the supply of (nutritious) foods within the urban context [24,25], in one case including demand for drivers for elderly age groups [26], no study was found that systematically assessed if spatial variation in demand, i.e., of dietary patterns, is detectable within the German context, or whether dietary patterns are spatially homogenous. 

Due to the limited understanding of spatial variation in dietary patterns in the German population, we aimed to better understand variations in food group intake and standard deviation scores of BMIs (BMI-SDS) of 6- to 18-year-old individuals in the Dortmund Nutritional and Anthropometric Longitudinally Designed (DONALD) cohort study. Therefore, we analyzed macronutrients and food groups, such as fruits and vegetables, as these food groups have been shown to reduce the risk of chronic disease, including in children and adolescents [27,28,29,30], and the consumption of sugar-sweetened beverages, as a higher intake of these is linked to a higher risk of a range of non-communicable diseases [31,32]. The objective of this study was to determine whether place of residence plays a role in the participants’ diets and BMI-SDS within the DONALD study.

## 2. Methods

### 2.1. Study Sample

The ongoing DONALD study is an open cohort study that was initiated in 1985 in Dortmund, the eighth largest city in Germany, located in the Ruhr area. The DONALD study’s design has been extensively described elsewhere [33]. In summary, each year, 35–40 healthy infants are recruited in or near Dortmund and undergo repeated examinations. Eligible are healthy German infants (i.e., infants free of diseases affecting growth and/or dietary intake) whose parents are willing to participate in a long-term study and of whom at least one has sufficient knowledge of the German language. Among 6–18 year olds, data on dietary intake, anthropometry, biomarkers, lifestyle, and early life parameters are collected each year. Parental examinations occur every four years. The DONALD study was approved by the Ethics Committee of the University of Bonn, according to the Declaration of Helsinki. Written consent of the study participants and/or parents was obtained prior to all investigations.

### 2.2. Study Population

For the present analysis, data of participants of the DONALD study, who were 6–18 years old between 2014–2019, were examined. The age range 6–18 was set to align with school-going age in Germany, meaning that these individuals are, to some degree, making individual decisions regarding their dietary intake. We selected the five years prior to the COVID-19 pandemic to exclude dietary intake that was potentially affected by national lockdowns, school closures, and an overall change in the food environment. We only selected participants with at least one anthropometric measurement.

In total, 360 participants who matched those requirements were identified. For dietary records, only 292 of those 360 participants had at least one observation and were included in the analysis of dietary intake. The median number of dietary records available for the 5-year period per participant was 3 (25. percentile: 2; 75. percentile: 5).

### 2.3. Dietary Assessment

Dietary intake information in the DONALD study is based on 3-day weighed dietary records. All food and drinks consumed by the participants, including leftovers, were weighed and recorded by the parents or the participants themselves, if old enough. Semi-quantitative recording (spoons, cups) was allowed if accurate weighing was not possible. Information on recipes, brands, and types of commercial foods was also required. Energy and nutrient intake were calculated based on the food composition database LEBTAB (LEBensmittelTABelle). Composition of staple foods is based on the German food composition table BLS (Bundeslebensmittelschlüssel) 3.02. Energy and nutrient contents of commercial food products were estimated by recipe simulation using labeled ingredients and nutrient contents. Commodity-level information was aggregated to food group level and standardized relative to individual energy intake (in g food group per 1000 kcal). Total energy intake (TEI), macronutrients (calculated as % of TEI), and food group intake were calculated as individual means of three days of recording. Food groups analyzed include grains, dairy, meat and fish, vegetables, fruits, sweets, and sugar-sweetened beverages (SSB) (Table 1).

### 2.4. Anthropometrics

For the DONALD study, height and weight were measured by trained nurses according to standard procedures, with the participants dressed in underwear only and barefoot. Standing height was measured to the nearest 0.1 cm using a digital stadiometer (Harpenden, Crymych, UK). Bodyweight was measured to the nearest 100 g using an electronic scale (Seca 753E; Seca Weighing and Measuring System, Hamburg, Germany). BMI was calculated using body weight (kg) divided by the square of the body height (m^2^). For an age- and sex-independent consideration of BMI, we calculated BMI–SDS using German reference percentiles for children and adolescents [34].

### 2.5. Place of Residence

Using GIS mapping, we assigned each individual reported place of residence to the respective administrative districts of Dortmund (Stadtbezirke). As the sample sizes in the northern districts were very small, we further grouped our data following a binary north (*n* = 52 for individuals with at least one anthropometric measurement) and south (*n* = 308) divide (Figure 1) to serve as proxy for neighborhood socio-economic status. We selected north/south rather than east/west as this divide aligns with reported differences in statistics on social status and wealth. These include unemployment, dependency on social assistance among the elderly, or enrolment in social protection scheme (SGB II, a social assistance mechanism by the federal employment agency, with the aim to cover subsistence costs for jobseekers and their dependents [35]): The latter being 34.7% in the north and 20.7% in the south for children, 21.2% in the north and 12.0% in the south for adults, respectively [36].

### 2.6. Assessment of Covariates

We included physical activity as covariate for BMI-SDS models to account for differences in energy balance [37]. We included socio-economic status of the household as covariate to account for differences that may influence the quality of the diet at household level [38]. Age and sex, taken from the annual participant questionnaire, were included to account for variation that may be due to lifecycle-specific eating habits [39]. Physical activity was expressed as Metabolic Equivalent of Task (MET)-minutes in organized, unorganized, and total settings. Data on physical activity were assessed using an interviewer-based, validated questionnaire (Adolescent Physical Activity Recall Questionnaire [40]), which included questions on duration and frequency of organized (e.g., training in a sports club) and unorganized (leisure sports, e.g., playing football with friends) activities.

Socio-economic data were collected using standardized questionnaires [32]. Socio-economic score (SES) was calculated adapted from [41], reflecting (a) educational and professional qualification of the parents and b) occupation level of parents, each receiving a score from 0–7, resulting in a total score from 0–14 per parent. No data on household income were available and, therefore, are not included in the calculation. Averages across both parents, where applicable, were calculated to estimate total household SES. Where only one score was available, this was taken as household SES.

### 2.7. Statistical Analysis

Data were processed, and descriptive statistics were prepared using RStudio 2022.07.1. All regression models were carried out using STATA version 16. The significance level was set at *p* < 0.05. A two-sample *t*-test with unequal variances was used to estimate whether a significant difference in SES exists between the two administrative groupings.

A robust multi-level mixed effects regression using STATA’s mixed command was used to analyze the effect of location of residence on nutrition and health outcomes. We define residence as the categorical independent variable and the dependent variable as per the following groups:BMI-SDSFood Group Dietary Intake (Grains, Vegetables, Fruits, Meat, Sweets, Dairy, SSB)Macronutrient Intake (Energy, Protein, Fat and Sugar).

Each indicator within these health and nutrition outcome groups was analyzed separately. We included individual’s unique ID and year as random effects to reflect varying numbers of observations per individual and changes due to specific years. We selected north as the reference area, i.e., coefficients are differences in the south relative to the north. For the first model (Model A), we included age (years) and sex (boy/girl) as covariables to account for the impact of these variables according to TEI. In the second model (Model B), we additionally included SES as a third covariable to account for possible effects that household SES may have (e.g., knowledge about healthy eating or resources available for food). For BMI-SDS as dependent outcome, we further included physical activity (measured in total MET-minutes) as a covariable (Model B*). We plotted and visually inspected histograms of residuals of the robust mixed effects model for normal distribution. We performed Breusch–Pagan/Cook–Weisberg test for heteroscedasticity using STATA’s hettest command and specified robust standard errors where heteroscedasticity is present (using STATA’s VCE command). Incomplete records were omitted from the respective models (Missing are *n* = 14 for SES-adjusted models and *n* = 3 for Activity-adjusted models). To account for multiple testing, Benjamini–Hochberg procedure for false discovery rate (FDR) was carried out within each model and outcome group, setting FDR at 0.20. The relatively high false discovery rate was selected due to the exploratory nature of this research and to avoid false negatives.

## 3. Results

Table 2 shows median values of background characteristics, anthropometric data, and dietary intake in this sample. SES is high overall in the study sample [33] but higher in the southern part of Dortmund than in the north. The range in SES, expressed by the 25th and 75th percentiles, was comparable in both areas, with the south having slightly higher values. Results from the *t*-test show that there is a small but significant difference in means of socio-economic status between the north relative to the south (diff = −0.70, *p* = 0.01).

In the fully-adjusted multi-level mixed-effects models using place of residence as the explanatory variable, we found that residence in the south was associated with lower BMI-SDS (β = −0.417, *p* = 0.017), lower intake of sugar-sweetened beverages (β = −47.00, *p* = 0.044) and higher intake of vegetables (β = 11.13, *p* = 0.043) in g per 1000 kcal after controlling for household SES (Table 3, unstandardized coefficients). No significant differences between north and south were found for the intake of fruit, meat, dairy, grains, or sweets. Additionally, no significant results were found for macronutrients, nor did adjustment for multiple testing change the significance of the findings (at FDR = 0.20, detailed results in Appendix A).

## 4. Discussion

This study investigated associations between place of residence and dietary intake and anthropometrics among children and adolescents enrolled in the DONALD study. This is the first time that data from this cohort were analyzed for spatial differences. Two main aspects stood out from the results: Firstly, we provided evidence that even when accounting for the overall high SES, spatial differences among participants of this rather homogeneous study population can be observed. Secondly, we presented an explorative analysis and starting point for further investigation into the enabling factors of the urban (food) environment in Germany. While spatial analysis of the urban food environment has been undertaken in several countries [3,42,43], no study could be identified that focused on the spatial differences in dietary intake within a German city or within the German population beyond urban/rural disaggregation. The association between SES and a healthy diet or health more generally has been documented for children and adolescents in the German context [41,44], with one study additionally finding that obesity contributes to the loss of SES [45]. The present analysis provided further nuance to the role of SES, for example, by distinguishing between neighborhood (spatial) and household levels of SES. We found that residence in the south, which is considered to have higher neighborhood socio-economic status [36,46], was significantly associated with lower BMI-SDS, lower intake of SSB, and higher intake of vegetables in the study population, even when accounting for household SES.

We observed a 0.417 difference in BMI-SDS between the north and south. With a completely random spatial distribution, the median value for both the north and south would be expected to be around 0 for each. However, predicted values for the south were −0.11 and for the north 0.30 for Model B*, respectively (a detailed overview of predicted values by the model is presented in the Appendix A). This pattern in BMI-SDS matched the general trends reported for overweight or obesity on the city district level among 6-year-olds in Dortmund [46].

For SSB, the estimated difference between the two regions was 47 g per 1000 kcal consumed—estimated consumption is around 65% higher in the north than it was in the south. Intakes of SSB reported in the KiGGS Study were at 300 mL per day for girls (11–13 and 14–17 years), nearly 330 mL per day for boys (11–13 years), and almost 500 mL per day for boys (14–17 years) [47]. The same study reported a significant discrepancy between socio-economic groups in the frequency of drinking SSBs. Although we did not analyze consumption frequencies in this paper, our data also show a higher intake of SSB in adolescent boys than in girls. The comparatively lower intakes per day reported in the DONALD study (compared to KiGGS, cf. Table 2) could be explained by the overall relatively high socio-economic status of the household. Despite the role that the SES of the household appears to play [48], differences between the two areas still occur when controlling for household SES. This indicated that factors beyond the household may also influence the intake of SSBs.

While the absolute differences between (adjusted) estimated intake of vegetables in north versus south Dortmund were small, estimates were 17% lower in the north. It is also noteworthy that the combined reported median quantity of fruits and vegetables consumed per day (123.85 g/1000 kcal for the north and 117.39 g/1000 kcal for the south) were far below the recommended daily intakes (roughly 190 g/1000 kcal, based on 400 g for both fruits and vegetables for the average individual (2100 kcal/day) according to the WHO). Hence, while the differences reported here were statistically significant, the actual difference between the two areas may be negligible, given that both areas would require large increases in intake to meet WHO targets.

As our results were adjusted for household socio-economic status in a spatial context, they, however, indicate the possibility that in some areas, even individuals that have relatively high levels of SES are limited in their ability to consume a healthy diet. Our findings, therefore, support the hypotheses that several determinants of dietary intake are outside the control of the individual [49]. Existing literature provides a variety of possible explanations why neighborhood socio-economic statuses could influence dietary patterns, including lack of availability [50,51,52,53], high relative prices of healthy foods [53], abundant and convenient access to unhealthy foods vis-à-vis healthier options [10,15], or peer effects within neighborhoods [54,55,56,57]. Further research is needed to identify specific drivers for the city of Dortmund, as well as the role of migration, which is higher in the north [36].

A growing body of research on the food environment exists, analyzing spatial variation in nutrition indicators generally [3,4,58,59], examining the influence of the food environment on dietary intake or health [8,60], the relationship between the socio-economic status of the neighborhood and food availability [15,51,52,53,61,62,63] and the influence of vendors on consumption [15]. Based on our data and analysis, our study belongs to the first group, as it focused on spatial variation in nutrition indicators. Specifically, our study added to the existing body of evidence on the food environment insofar as we showed (1) significant variation between two city areas, which aligned with global findings that highlight the role of the socio-economic status of the neighborhood, and (2) that the impact of individual or household socio-economic status may not be sufficiently strong to counteract the role that the food environment had.

While most research includes features of the built environment (such as store density or assortment), our study did not include such characteristics. Still, our findings are in line with evidence from other countries, which suggest that the physical area of residence may have a strong influence on eating patterns and, consequently, nutrition and health status, even when the background characteristics of the individual are accounted for. In a systematic review of 15 studies on obesity in children and adolescents and urban built environments carried out in the USA between 2001 and 2008, Dunton et al. reported that neighborhood pattern was the only feature that showed an association with obesity [9]. In another study conducted with data from Los Angeles, USA, researchers found that living in a neighborhood with a very low SES residence score was significantly associated with higher BMI, even when accounting for the education and income of the individual [60]. In a similar study focusing on dietary patterns of female adolescents in Baltimore, USA, Hager et al., argued that “without health-promoting opportunities from higher-ranking systems (i.e., the neighbourhood), individuals (i.e., adolescents) have difficulty pursuing and maintaining healthy behaviours” [15]. They found that neighborhood SES was associated with the consumption of snacks and desserts but not with the number of servings of fruits and vegetables of the individual. As they did not investigate the portion size of food groups, it remains unclear whether there was a (small) difference in intake amounts, as was found in our study.

Beyond its exploratory value, the present evaluation showed three specific strengths: Firstly, the longitudinal design: we accounted for dietary intake over 5 years and used multiple observations per individual (median *n* = 3). Secondly, we included a quantitative household SES indicator: we included information on educational and professional qualifications as well as the actual professional status of the parents. Thirdly, our dataset allowed us to analyze the weight of consumed food groups, whereas many studies focused on overweight, aggregated groups (“snacks and desserts”) or frequencies (“servings of fruit and vegetables”).

Yet, some empirical, methodological, and conceptual limitations remain: Despite variation in the group, the study sample still had relatively high socio-economic status (mean value 10 [from 0–14]), which has been noted and discussed elsewhere [33]. This analysis is, therefore, not generalizable beyond the study sample and only allows limited conclusions for a wider population. The calculated SES consisted only of two out of three indicator groups used by the reference literature: education and employment, not factoring in the actual income as this is not available in the dataset. It may, therefore, be possible for residual socio-economic confounding to occur. Although we used robust standard errors, accounted for possible confounders, and the fact that low numbers of observations in the north are generally reflected in wider confidence intervals, the uneven geographic distribution of participants may have impacted our findings. Furthermore, geographic location was included as a two-area variable (north/south) in the statistical models, which may not reflect the diversity of locations in Dortmund. It is, therefore, possible that statistically significant clusters exist that were not identified by this analysis and equally possible that clusters documented here are not visible in a different definition or aggregation of area units. Additionally, differences in genders and age groups, which are not spread equally across the two areas, could also have influenced findings. However, we did account for these parameters statistically by including age and sex as covariates. Lastly, we did not provide any evidence on how the reported differences could arise. We documented that there is significant spatial variation in dietary intake and nutrition status between the two areas, but investigation of why was beyond the scope of this analysis.

## 5. Conclusions

To our knowledge, this is the first study that looked at the association between nutrition indicators (dietary intake and BMI-SDS) and the location of inner-city residences in Germany. We found that even when adjusting for age, sex, socio-economic status (and physical activity for BMI-SDS), there was a significant difference between the north and the south of Dortmund for BMI-SDS and intake of SSB and vegetables. Results for grains, dairy, animal protein sources, fruits, or sweets are statistically insignificant and therefore warrant further investigation.

Overall, our results add to existing evidence that factors affecting dietary intake are in part outside of the immediate control of the individual. Further research is therefore required to arrive at conclusive evidence on whether place of residence is a significant variable in determining dietary intake or whether inner-city spatial clusters are driven by individual or household socio-economic factors alone. This should include both further (spatial) analysis of existing datasets and the collection of primary data on the urban food environment itself.

## Figures and Tables

**Figure 1 ijerph-21-00046-f001:**
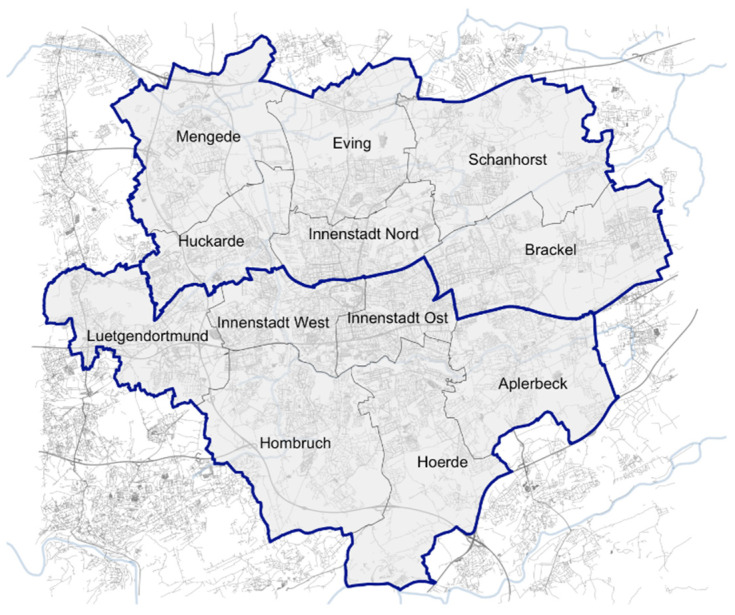
Dortmund city districts per analysis area (north and south).

**Table 1 ijerph-21-00046-t001:** Description and components of food groups utilized for this analysis.

Food Group	Components
Meat and Fish	Beef, veal, pork, game, lamb, goat, horse Poultry Organ meats and offal Sausages, cold cuts Meat dishes Fish, fresh or frozen Processed fish
Dairy	Dairy products, fermented and unfermented Fresh, soft, (semi)hard and processed cheese Dairy powder Instant milk beverages (e.g., cocoa)
Fruits	Fruit, fresh and frozen Fruit, canned Fruit, dried
Vegetables	Vegetables and mushrooms, fresh and frozen Vegetables and mushrooms, canned Vegetables and mushrooms, dried
Sweets	Sugar, sweeteners Sweet parfait (jam, honey, hazelnut spread) Candy (wine gums, drops) Chocolate, bars Ice cream, water ice Sweet sauces Non-milk-desserts
Grain	Flour, mixed and plain Bread Grain, cooked and raw Dough Pasta Ready to eat cereals
SSB	Sweetened fruit juice drinks and nectars, Soft drinks/sodas, Sweetened teas and waters, Instant beverages (except dairy drinks), sweetened sport drinks

**Table 2 ijerph-21-00046-t002:** Characteristics of DONALD study participants (age 6–18) between 2014 and 2019, stratified by place of residence.

Place of Residence:		North		South	
*n* participants		52	(14.4)	308	(85.6)
n_anthropometry_ ^a^		184	(14.5)	1083	(85.5)
n_3-day-dietary-records_ ^a^		149	(15.9)	786	(84.1)
*n* female participants (%)		27	(51.9)	134	(43.5)
Age (in years)		14	(9.8; 16.5)	11	(7.6; 15.1)
** *SES ^b^* **					
	Household SES	9.4	(7.8; 10.8)	10.3	(9.1, 11.3)
** *Anthropometry* **					
	BMI (kg/m^2^)	19.8	(16.7; 23.4)	17.4	(15.5; 20.4)
	BMI-SDS	0.4	(−0.6; 1.2)	−0.2	(−0.8; 0.4)
	Height (cm)	164.9	(139.6; 172.9)	148.7	(129.1; 168.0)
	Weight (kg)	52.3	(33.7; 67.9)	38.1	(26.8; 57.9)
** *Macronutrients ^c^* **					
	TEI (kcal/day)	1795.2	(1519.8; 2057.8)	1694.6	(1481.4; 1978.0)
	Carbohydrates (%E)	50.6	(47.5; 53.2)	51.1	(47.5; 54.2)
	Fat (%E)	33.6	(32.2; 37.7)	34.4	(31.9; 37.7)
	Protein (%E)	13.2	(12.5; 14.8)	13.4	(11.8; 14.9)
	Sugar (%E)	23.8	(20.1; 26.3)	22	(18.8; 26.0)
** *Food Group ^c^* **					
	Dairy (g/1000 kcal)	136.1	(102.6; 188.5)	124.4	(88.5; 182.6)
	Fruit (g/1000 kcal)	59.7	(25.9; 92.5)	61.1	(37.7; 97.6)
	Grains (g/1000 kcal)	85.4	(67.2; 96.9)	83.7	(64.4; 103.1)
	Meat and Fish (g/1000 kcal)	55.9	(42.2; 72.0)	46.1	(31.8; 69.2)
	SSB (g/1000 kcal)	68.5	(17.4; 158.3)	39.5	(6.5; 80.8)
	Sweets (g/1000 kcal)	29.1	(23.1; 42.2)	33.3	(19.7; 47.1)
	Vegetables (g/1000 kcal)	64.1	(38.0; 83.1)	56.3	(33.5; 84.2)
** *Physical Activity ^d^* **					
	MET Minutes	967.8	(560.9; 1304.1)	1014	(708.4; 1427.0)

Values are medians (25th; 75th percentile in parenthesis) of the mean of repeated measurements by participant, or frequencies (% in parenthesis). SES Socio-Economic Score, SSB Sugar-Sweetened Beverages, BMI Body Mass Index, BMI-SDS Body Mass Index Standard Deviation Score, TEI total energy intake, MET Metabolic Equivalent of Task. ^a^ Due to repeated measurements per participant. ^b^ Missing 14 (participants), north = 3, south = 11. ^c^ Missing 68 (participants), north = 6, south = 62. ^d^ Missing 3 (participants), north = 0, south = 3.

**Table 3 ijerph-21-00046-t003:** Spatial trends between south and north (reference) Dortmund for selected indicators.

			β	*p*-Value	Lower CI	Upper CI
**BMI**	SDS	Model A	−0.489	**0.005**	−0.827	−0.151
		Model B	−0.425	**0.016**	−0.769	−0.081
		Model B*	−0.417	**0.017**	−0.759	−0.075
**Food Groups**	SSB	Model A	−47.661	**0.039**	−92.821	−2.501
		Model B	−47.000	**0.044**	−92.642	−1.359
	Vegetables	Model A	12.133	**0.027**	1.377	22.889
		Model B	11.129	**0.043**	0.351	21.906
	Fruit	Model A	7.829	0.302	−7.030	22.688
		Model B	9.876	0.196	−5.094	24.847
	Meat	Model A	−4.468	0.252	−12.122	3.185
		Model B	−4.746	0.239	−12.640	3.148
	Sweets	Model A	1.602	0.545	−3.592	6.796
		Model B	2.327	0.397	−3.057	7.712
	Grain	Model A	3.905	0.277	−3.141	10.952
		Model B	1.646	0.637	−5.200	8.492
	Dairy	Model A	−8.138	0.465	−29.946	13.671
		Model B	−6.237	0.565	−27.492	15.017

Significant *p*-values (*p* < 0.05) in the adjusted model are bolded; Model A is adjusted for age and sex. Model B is adjusted for age, sex and socio-economic status. Model B* (BMI-SDS only) is adjusted for age, sex, socio-economic status and physical activity. β-coefficients are unstandardized. BMI Body Mass Index, SDS Standard Deviation Score, SSB Sugar-Sweetened Beverages, CI Confidence Intervall.

## Data Availability

Data of the DONALD study is available upon request.

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
