# Peer review of "Place of Residence Is Associated with Dietary Intake and BMI-SDS in Children and Adolescents: Findings from the DONALD Cohort Study"

_ijerph, 2023, doi:10.3390/ijerph21010046_

Round 1

Reviewer 1 Report

Comments and Suggestions for Authors

It was a great pleasure that I reviewed the manuscript entitled “Place of residence is associated with dietary intake and BMI-SDS in children and adolescents: Findings from the DONALD cohort study.” I think the paper presents interesting findings. Here are several comments and suggestions that might help strengthen the paper. I am presenting those comments and suggestions in (mostly) chronological order.

1.     I think the Introduction section is overall well-written. However, the authors need to discuss more details in the section of 2.5 Place of residence. For example, the authors should provide more information what enrollment in social-protection schema means.  It sounds like the residents in south are wealthier, but is there other measure to indicate this?

2.     It is my understanding that the authors tested to see the effect of the residence on the dependent variables were observed after controlling for SES in Model B. Thus, primary findings reported in the results section should highlight the fact that the effects were all observed after SES is controlling for.

3.     Nevertheless, the way the authors discussed in the discussion section is a little confusing. The authors stated that “We found that residence in the south, which is considered to be of higher socio-economic status, is significantly associated with lower BMI-SDS …”, which should be true (though I cannot tell this effect from the table), but I think this is just mudding the waters.

4.     More generally, if my interpretation above is correct, I think the authors should address the fact that the effect of the south is a significant predictor for the dependent variables after controlling for SES by discussing potential reasons. If there are other reasons beside being wealthy for healthy, then this study simply provides the relatively new but very limited findings unless discussing them.

Minor points:

5.     On Page 5, consider the use of BMI-SDS all the time.

6.     Please clarify if the authors reported whether unstandardized or standardized beta.

Reviewer 2 Report

Comments and Suggestions for Authors

Dear Authors,

Thanks for the study “Place of residence is associated with dietary intake and BMI-SDS in children and adolescents: findings from the DONALD cohort study”. The aim of study was to determine whether location of residence in the German urban food environment is associated with habitual dietary intake and body mass index.

The publication needs some revisions:

1.      The introduction of the publication lacks a review of previous publications on the chosen topic in other countries.

2.      There is a missing research question or hypothesis that could be put forward if the results of similar studies were expanded in the introduction.

3. The publication needs to explain the results in more detail. For example, the northern part has more boys and the average age is higher, which means higher weight and height. All this affects the results in terms of eating habits. There are differences in eating habits between genders and also age groups, which may have influenced the fact that the consumption of sugar sweetened beverages is higher in the north. It is strongly recommended to expand the analysis of the results in discussion and look at the influence of the various factors and provide an explanation.

Round 2

Reviewer 2 Report

Comments and Suggestions for Authors

Dear authors,

Thank you for improving the manuscript. In my view, the publication can be published in a journal.

Good luck with your future research.